# Neural control of lexical tone production in human laryngeal motor cortex

Junfeng Lu [1,2,3,9], Yuanning Li [4,5,6,7,9], Zehao Zhao [1,2,3,9], Yan Liu[1,2,3], Yanming Zhu [1,8], Ying Mao [1,2,3] ✉, Jinsong Wu [1,2,3] ✉ & Edward F. Chang [5,6] ✉

In tonal languages, which are spoken by nearly one-third of the world's population, speakers precisely control the tension of vocal folds in the larynx to modulate pitch in order to distinguish words with completely different meanings. The specific pitch trajectories for a given tonal language are called lexical tones. Here, we used high-density direct cortical recordings to determine the neural basis of lexical tone production in native Mandarin-speaking participants. We found that instead of a tone category-selective coding, local populations in the bilateral laryngeal motor cortex (LMC) encode articulatory kinematic information to generate the pitch dynamics of lexical tones. Using a computational model of tone production, we discovered two distinct patterns of population activity in LMC commanding pitch rising and lowering. Finally, we showed that direct electrocortical stimulation of different local populations in LMC evoked pitch rising and lowering during tone production, respectively. Together, these results reveal the neural basis of vocal pitch control of lexical tones in tonal languages.

In spoken languages, vocal pitch is an important acoustic cue for both lexical and non-lexical information. In non-tonal languages, such as English, vocal pitch represents prosody and intonation[1]. In tonal languages, in contrast, vocal pitch is mainly used to distinguish single words from each other. In Mandarin Chinese, for example, a single syllable can feature one of four different pitch contour categories (lexical tones) to signify different Chinese characters and meanings[2].

Tone categories are usually classified by two features: the starting pitch height (high, mid, or low) and the direction of change (rising, falling, or dipping). For example, in Mandarin, when the syllable /ma/ is pronounced with a high pitch (high-level tone, /mā/, Tone 1), it means mother (妈). When the pitch contour rises during the syllable (mid-rising tone, /má/, Tone 2), it means hemp (麻). When the pitch drops

and then increases (low-dipping tone, /mǎ/, Tone 3), it means horse (马). Finally, a falling tone (high-falling tone, /mà/, Tone 4) means to scold (骂)[3]. Therefore, precise control of the larynx to produce vocal pitch across a broad dynamic range at the timescale of single syllables is critical for tonal language speakers.

Three key functions of the larynx are involved in the production and modulations of pitch: voicing, pitch rising, and pitch lowering (Fig. 1a)[4]. Voicing, also known as vocalization or phonation, is created by the muscles in the larynx bringing the vocal cords together. As the air rushes through the vocal tract, the ligament of the vocal cords vibrates passively, producing the fundamental frequency ($F_0$), which is determined by the physical properties of the vocal folds[4,5]. Two major intrinsic laryngeal muscles, namely the cricothyroid (CT) and

[1]Department of Neurosurgery, Huashan Hospital, Shanghai Medical College, Fudan University, Shanghai 200040, China. [2]Shanghai Key Laboratory of Brain Function Restoration and Neural Regeneration, Shanghai 200040, China. [3]National Center for Neurological Disorders, Huashan Hospital, Shanghai Medical College, Fudan University, Shanghai 200040, China. [4]School of Biomedical Engineering, ShanghaiTech University, Shanghai 201210, China. [5]Department of Neurological Surgery, University of California, San Francisco, CA 94143, USA. [6]Weill Institute for Neurosciences, University of California, San Francisco, CA 94158, USA. [7]State Key Laboratory of Advanced Medical Materials and Devices, ShanghaiTech University, Shanghai 201210, China. [8]Speech and Hearing Bioscience & Technology Program, Division of Medical Sciences, Harvard University, Boston, MA 02215, USA. [9]These authors contributed equally: Junfeng Lu, Yuanning Li, Zehao Zhao. ✉e-mail: maoying@fudan.edu.cn; wujinsong@huashan.org.cn; edward.chang@ucsf.edu

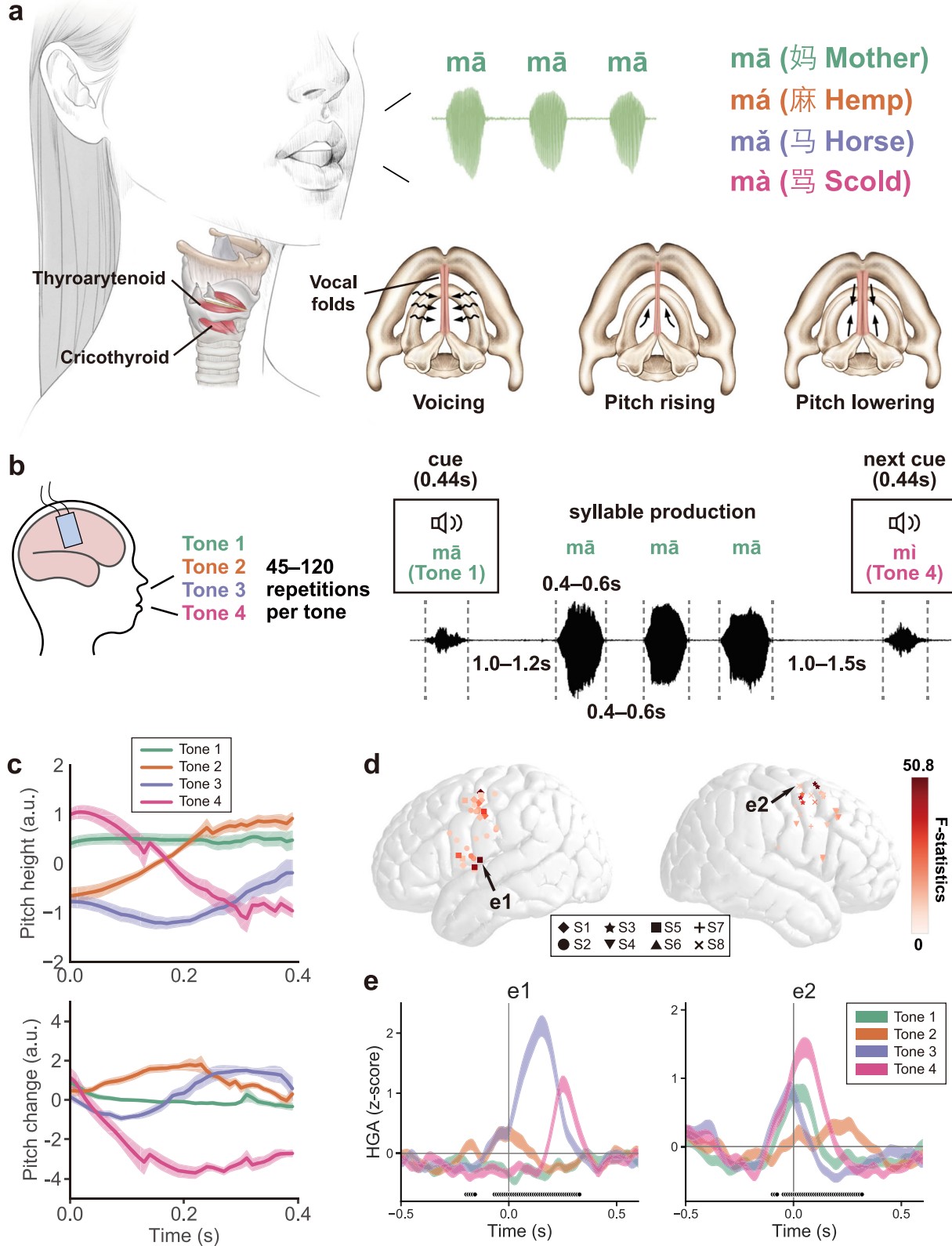

thyroarytenoid (TA) muscles, achieve fine control of the tension of vocal folds. In the process of voicing, CT stretches and increases the tension of vocal folds to raise the vocal pitch, while TA shortens and decreases the tension of vocal folds with other muscles to lower the pitch[4,6,7].

Neuroimaging and neurophysiological studies have recently identified two laryngeal motor cortex (LMC) regions in the human

ventral sensorimotor cortex (vSMC) correlated with laryngeal movements[8–12]. The bilateral dorsal LMC (dLMC) were found to monotonically encode pitch rising during a word emphasis task in English[8]. Voicing is encoded by distinct neural populations in both dLMC and ventral LMC (vLMC)[8,13].

However, how humans exert precise control of laryngeal muscles to dynamically regulate vocal pitch to produce lexical tones is still

**Fig. 1 | Differential neural activity patterns are represented in single electrodes from LMC during lexical tone production. a** Anatomical and physiological basis of vocal pitch control (copyright Jinxuan Wu, used with permission). When the subject is uttering syllables of four different Mandarin tones, the vocal cords are pushed together and vibrated by the passing air to produce the sound (voicing), cricothyroid stretches and increases the tension of vocal folds to raise the pitch, while thyroarytenoid contracts and decreases the tension of vocal folds to lower the pitch. **b** The flowchart of the lexical tone production task (copyright Zehao Zhao, used with permission). In each trial, the participants were presented with a randomly displayed tonal syllable as an auditory cue (0.44 s). Following a reaction time interval (1.0–1.2 s), the patients were instructed to repeat the word three times consecutively (0.4–0.6 s per word) with intervals of 0.4–0.6 s between repetitions. The next auditory cue was played after 1.0–1.5 s. Each patient completed 60-160 trials, resulting in 45–120 repetitions per lexical tone. **c** The average pitch dynamics (arbitrary units [a.u.]) for four lexical tones produced by all subjects. Upper panel: the average trajectory of pitch height (solid line: mean, shaded area: standard errors of the mean [s.e.m.] across all repetitions for each tone of all subjects); lower panel: the average trajectory of pitch change (solid line: mean, shaded area: s.e.m. across all repetitions for each tone of all subjects). The 0 ms time point represents the onset of the vowel. **d** The distribution of all tone discriminant electrodes. Different symbols represent electrodes from different subjects (S1-S8). Color bar indicates the degree of tone discrimination (F-statistics across 4 tones). **e** The average high-gamma responses for different lexical tone production (shaded area represents mean ± s.e.m. across repetitions of each tone) in two example electrodes (locations indicated by black arrows in (**c**). The 0 ms time point represents the onset of the vowel. Black dots indicate time points of significant difference across 4 tones ($p < 0.05$, one-way ANOVA, Bonferroni corrected). Source data are provided as a Source Data file.

unanswered. A positive monotonic coding for pitch rising cannot fully account for the pitch lowering dynamics in tonal languages. A previous intracranial study[14] in Mandarin tone perception demonstrated the existence of single-electrode-level high gamma responses that differentiate between lexical tones in the temporal lobe during speech perception. Such neural activity can be explained by positive and negative tuning to speaker-normalized pitch features. Here, we seek to determine the neural coding mechanisms underlying lexical tone production. Specifically, if it is represented by the encoding of laryngeal articulatory kinematics/pitch dynamics or more abstract planning signals of discrete tone categories. We also want to understand which pitch parameters, and specifically, whether the pitch height or the more complex encoding of the underlying pitch modulation functions (pitch rising and pitch lowering) are encoded during tone production. Last, we want to determine whether the neural coding of pitch is localized in specific cortical areas or distributed in vSMC.

To address these questions, we used high-density intracranial recordings from participants undergoing neurosurgical brain mapping procedures. We recorded neural activity from the ventral sensorimotor cortex while eight participants spoke Mandarin syllables with four different tones. The wide range and change of pitch in tonal language produced by laryngeal muscles, in combination with high spatial and temporal resolution recordings allow us to investigate the neural encoding of vocal pitch. Finally, we used direct cortical electrical stimulation to probe the exact causal relationship between the specific neural population and the corresponding pitch dynamics.

## Results

### Differential neural activity patterns are represented in single electrodes from LMC during tone production

High-density electrocorticography (ECoG) arrays were placed temporarily over the lateral sensorimotor cortex (left $n = 4$, right $n = 4$). Participants spoke into a microphone while simultaneous neurophysiological ECoG recordings were done. In each trial of the experiment, the participants randomly heard a monosyllabic word (e.g., /ma/, /mi/) with one of the four different tones, and were instructed to repeat the monosyllable aloud three times in a row (Fig. 1b). Each participant completed 60-160 trials, which yielded 45-120 repetitions per lexical tone. Pitch height and pitch change are two critical pitch features that were shown to be important in discriminating lexical tones in tone languages[2,3]. We extracted the pitch contour ($F_0$) from the produced acoustic waveform and examined these two features for all participants. Robust and discriminable contours for different tones were produced across all 8 participants, spanning a broad range in both pitch height and pitch change (Fig. 1c). Additionally, a principal component analysis (PCA) of the pitch contours of tone tokens from all participants also revealed similar key features (Fig. S1).

We computed the analytic amplitude of signals in the high-gamma band (70 to 150 Hz), a measure correlated with local neuronal activity[8,15]. We found widespread cortical activation in the vSMC during syllable articulation, with 56.0 (±18.5 s.d.) active electrodes per participant on average (see Fig. S2 for the distribution of speech responsive electrodes on individual surfaces).

We then aligned the high-gamma activity for each tone token to the vowel onset time, and evaluated the average neural response for each tone across syllable tokens. Electrodes in bilateral dLMC and left vLMC showed differential patterns of neural activity to the four tones during articulation ($p < 0.05$, one-way ANOVA, Bonferroni corrected; Fig. 1d). These electrode sites with tone discriminating activity made up an average of 14.2% (±9.7% s.d.) of all articulation-related sites. Furthermore, among these tone discriminant electrodes different response profiles were found, where each electrode was activated during the production of multiple tones, and each tone elicited different patterns of neural activity across different electrodes (Fig. 1e). Therefore, the results do not show evidence of electrodes sites that were tuned to a single tone category but rather suggest a distributed neural coding in bilateral LMC underlying the cortical control of lexical tone production.

### Cortical patterns of tone discriminant electrodes are explained by pitch height and change encoding

Since we did not find tone-specific coding in the tone discriminant electrodes, we next investigated what features drive the differential patterns in these electrodes. To do this, we used a linear encoding model to predict the neural activity during tone production, using features representing the pitch dynamics (pitch height and pitch change), voicing (binary pitch), non-laryngeal articulator movements and so on. The unique contribution of each feature was computed and correlated to the tone discriminability in the neural activity for each electrode ($n = 448$). A strong and significant positive correlation was found between pitch encoding and tone discriminability, suggesting that the differential patterns of neural activity were mainly driven by pitch encoding (Pearson's $r = 0.75$, $p = 4.3E{-}36$; Fig. 2a). No correlation was found between tone discriminability and intensity (Pearson's $r = 0.05$, $p = 0.37$; Fig. 2b), syllable onset (Pearson's $r = 0.05$, $p = 0.40$; Fig. 2c), binary pitch (Pearson's $r = 0.03$, $p = 0.67$; Fig. 2d) or tone category (Pearson's $r = 0.07$, $p = 0.25$; Fig. 2e). We further looked at the two components of pitch features. We found neural activity at individual electrodes either represents pitch height or pitch change (Fig. 2f). Pitch change was more relevant to tone discriminability than pitch height (Pearson's $r = 0.61$ for pitch change, Pearson's $r = 0.45$ for pitch height, $z = 64.3$, Fisher's $z$ transform comparing two correlation coefficients, $p < 1E{-}100$; Fig. 2g, h).

Since the pitch contours of lexical tones in continuous natural speech might deviate from those in isolated canonical single tonal syllables, we next wanted to evaluate whether findings in the syllable production task would generalize to speaking natural sentences as well. Participants were asked to speak aloud 20 phonetically balanced Mandarin sentences[16], reading from a prompt on a screen. We found the neural activity to lexical tones in natural speech was

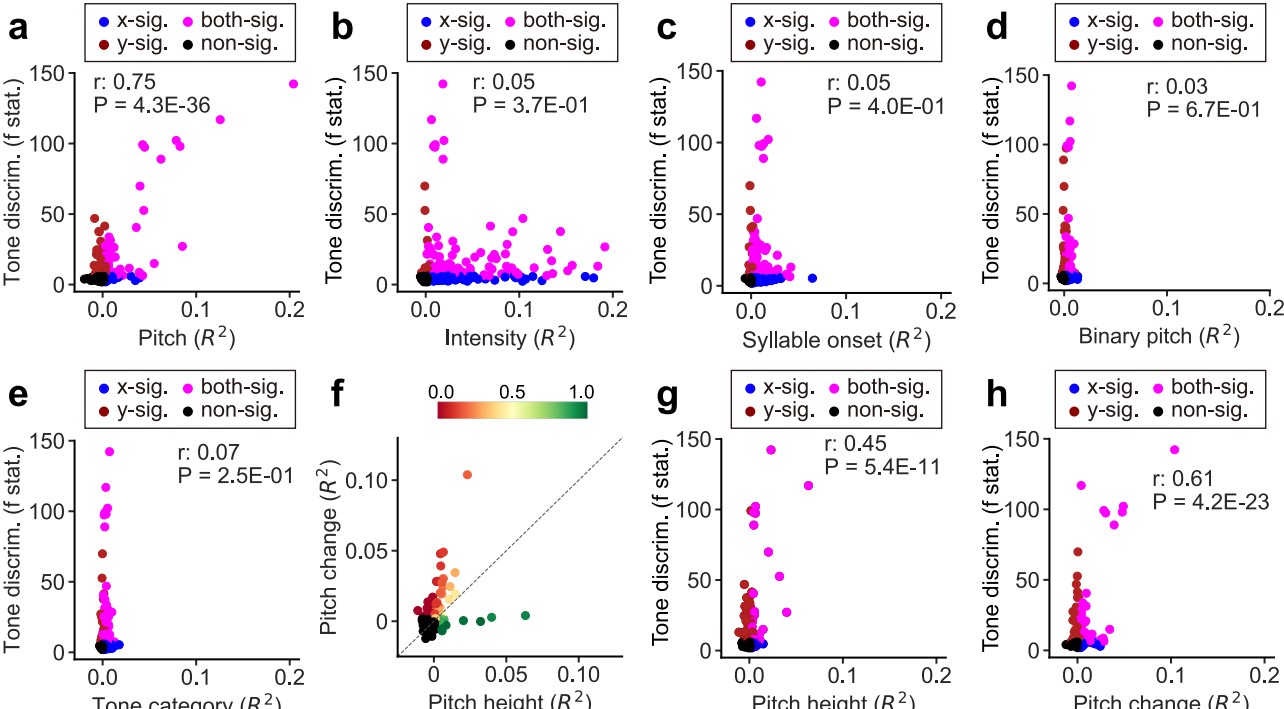

**Fig. 2 | Tone discriminability from the neural activity is correlated with pitch height and pitch change representation. a–e, g, h** Scatter plot of single electrode encoding properties across speech-selective electrodes from all participants. Each dot represents a single electrode ($n = 448$). The colored dots indicate electrodes that had significant encoding of $x$-axis (blue, $p < 0.005$, one-sided permutation test), $y$-axis (red, $p < 0.05$, one-way ANOVA, Bonferroni corrected) encoding features or both (magenta), while the black dots indicate non-significant electrodes. The $r$ and $P$ values are computed among the significant electrodes (colored dots) using Pearson's correlation (two-sided) between the $x$- and $y$-axis. **a** Scatterplot of the unique variance explained by pitch features ($R^2$ of pitch change and pitch height) and tone discriminability ($F$-statistics across 4 tones). **b** Scatterplot of the unique variance explained by intensity ($R^2$) and tone discriminability ($F$-statistics).

**c** Scatterplot of the unique variance explained by syllable onsets ($R^2$) and tone discriminability ($F$-statistics). **d** Scatterplot of the unique variance explained by binary pitch feature ($R^2$) and tone discriminability ($F$-statistics). **e** Scatterplot of the unique variance explained by tone category ($R^2$) and tone discriminability ($F$-statistics). **f** Scatterplot of the unique variance explained by pitch change ($R^2$) and pitch height ($R^2$). The colored dots indicate electrodes that had significant encoding of either pitch change or pitch height ($p < 0.005$, one-sided permutation test), with color indicating the proportion of variance explained by pitch height relative to the total variance explained by pitch change and pitch height. **g** Scatterplot of the unique variance explained by pitch height ($R^2$) and tone discriminability ($F$-statistics). **h** Scatterplot of the unique variance explained by pitch change ($R^2$) and tone discriminability ($F$-statistics). Source data are provided as a Source Data file.

still best explained by the pitch features (Fig. S3), and the single electrode encoding properties were largely consistent between the syllable production and natural sentence production tasks (Fig. S4). In brief, the neural activity of tone-discriminating electrodes represented the pitch dynamics during both syllable and sentence production.

## Control of vocal pitch is coordinated by two distinct tuning patterns in LMC

We next wanted to understand mechanistically how these tone-encoding electrodes generate pitch dynamics. The pitch contour is directly related to the movements of two groups of laryngeal muscles. In non-tonal languages, such as English, most of the pitch dynamics involves pitch rising above the neutral point, which is created by the stretchy muscles. However, in tonal languages, such as Mandarin, pitch dynamics also involves pitch lowering below the neutral point, which is created by the muscles that thicken and shorten vocal folds. To model these two distinct patterns of pitch dynamics and muscle movements, we adapted the computational pitch contour generation model for Mandarin, called the Fujisaki model[17]. The Fujisaki Model can simulate the physiological operation of the human larynx, which enables us to decompose Mandarin pitch contour ($F_0$) into tone command, phrase command, amplitude, and timing components, and to use variations in these components to synthesize different tones. Using the Fujisaki model, the rapid tone commands were extracted to generate pitch contours for different tones. For each tone, the tone commands can be

both positive and negative. Specifically, Tone 1 can be synthesized by a long positive tone command; the rising Tone 2 requires an early negative and a late positive tone command; a long negative command generates the low-dipping Tone 3; and the falling Tone 4 is combined with an early high positive and a late negative command. Using these extracted tone commands, the dynamic pitch contours of each tone were reconstructed with high fidelity (Fig. 3a), confirming the performance of the Fujisaki model for the participants' utterances.

An encoding model, similar to the one described in the previous section and with the extracted tone commands, was used to predict the high-gamma activity with regard to tone production in each individual electrode. The differential neural activity for tones in single electrodes was also well explained by the tone commands (Pearson's $r = 0.72$, $p = 7.9E-41$; Fig. 3b). To further understand the neural coding mechanism at the level of a single electrode, we computed the tuning curve for each individual electrode, i.e., the averaged high-gamma activity as a function of the tone commands. We found distinct tuning patterns at different electrodes. Specifically, we found electrodes with positive tuning of tone commands (22/54 of total electrodes), and electrodes with negative tuning of tone commands (32/54 of total electrodes), both distributed bilaterally in LMC (Fig. 3c, d). Moreover, the tone control electrodes were clustered in the bilateral dLMC (16/54 of total electrodes in left dLMC, 21/54 of total electrodes in right dLMC) and the left vLMC (14/54 of total electrodes) (Fig. 3e, see also Fig. S5 for the distribution of tuning electrodes on individual surfaces). Together, these results suggest that bilateral LMC has a distributed

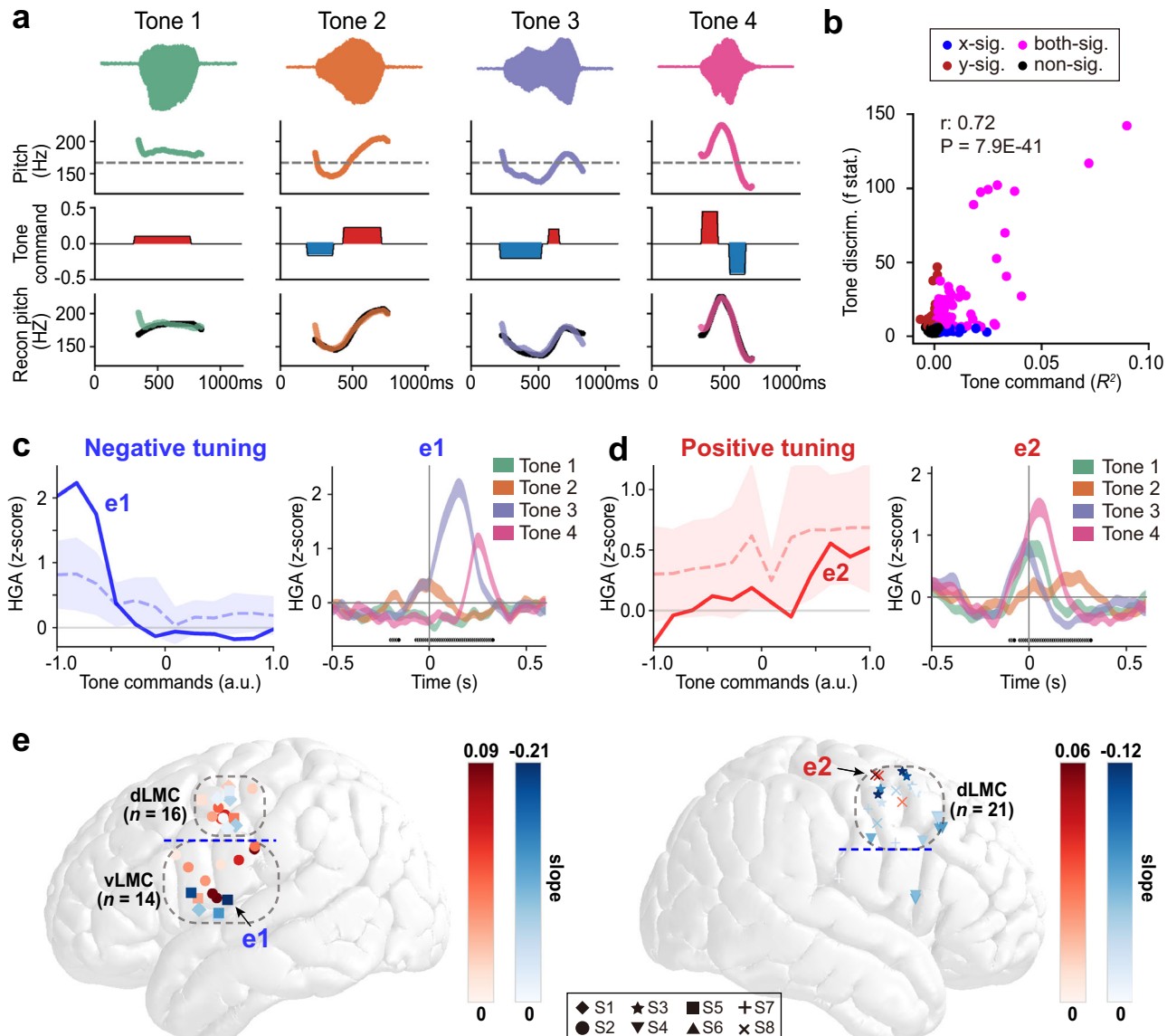

**Fig. 3 | Two tuning commands exist in the LMC to generate different tone control of vocal pitch. a** The application of Fujisaki model. First row: example syllable waveforms of different tones; second row: original pitch trajectories; third row: positive (red) and negative (blue) tone commands extracted using Fujisaki model; fourth row: reconstructed pitch trajectories using Fujisaki model (black curve), compared to original trajectories. **b** Scatter plot of single electrode encoding of tone command features ($R^2$) versus tone discrimination ($F$-statistics) across speech-selective electrodes from all participants. Red dots indicate electrodes that had significant tone discrimination only ($p < 0.05$, one-way ANOVA, Bonferroni corrected); Blue dots indicate electrodes that had significant encoding of tone command only ($p < 0.005$, one-sided permutation test); Magenta dots indicate electrodes that had significant encoding of both tone discrimination and tone command. The $r$ and $P$ values are computed among significant electrodes (colored dots) using Pearson's correlation. **c** Left: average neural tuning curve for negative tuning electrodes (dashed line: mean high-gamma response, shaded area:

s.e.m.). Solid curve corresponds to an example electrode e1. Right: average high-gamma activity for different tones (shaded area represents mean ± s.e.m. across repetitions of each tone) in electrode e1. **d** Left: average neural tuning curve for positive tuning electrodes (dashed line: mean high-gamma response, shaded area: s.e.m.). Solid curve corresponds to an example electrode e2. Right: average high-gamma activity for different tones (shaded area represents mean ± s.e.m. across repetitions of each tone) in electrode e2. In (**c**, **d**), 0-ms time point represents the vowel onset. Black dots indicate time points of significant difference across 4 tones ($p < 0.05$, one-way ANOVA, Bonferroni corrected). **e** Distribution of positive (red) and negative (blue) tuning electrodes on the cortical surface. Different symbols represent significant tone tuning electrodes from different subjects. Darker color indicates stronger tuning. The example electrodes e1 and e2 are marked by black arrows. Blue dashed line represents the boundary line separating the dLMC and vLMC clusters (indicated by gray dashed circles). See also Fig. S5. Source data are provided as a Source Data file.

neural coding of tone commands, where different local populations show different patterns of encoding for positive and negative tone commands.

We have demonstrated neural encoding at individual electrodes. Subsequently, to determine how neural responses to different tones were spatiotemporally distinguishable in the distributed LMC network, we used multivariate pattern analysis to evaluate the tone decoding accuracy in LMC neural population. We collected all

speech-responsive electrodes across all subjects and trained pairwise classifiers to decode the 4 lexical tones from each other. We found that tones could be significantly decoded as early as 300 ms before the pitch onset (Fig. 4a). The population decoding accuracy peaked at 66.7%, and around 250–300 ms after vowel onset (Fig. 4a). Consistent with the encoding analysis, electrodes that contributed to the tone discrimination distributed in bilateral dLMC and left vLMC (Fig. 4b).

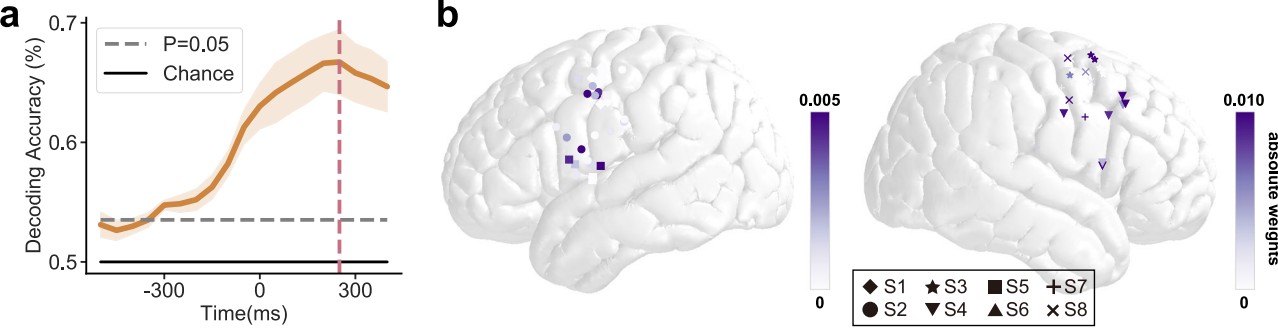

**Fig. 4 | Tone can be decoded from distributed neural activity. a** Time course of mean pair-wise tone classification accuracy using a sliding time window of 100 ms (solid line: mean, shaded area: s.e.m. across subjects S1-S8) and population neural activity in all speech-responsive electrodes in bilateral LMC. The 0 ms time point represents the onset of the vowel. Horizontal dashed line: $p = 0.05$ threshold, two-sided t-test for independent samples, Bonferroni corrected; vertical dashed line: peak accuracy time. **b** Averaged classifier weights at peak accuracy time on each electrode. Each dot represents one electrode (different symbols represent electrodes from different subjects), with a darker color indicating a larger absolute classifier weight. Source data are provided as a Source Data file.

## Intraoperative direct electrical stimulation of dLMC elicits pitch rising and lowering

We have demonstrated the neural activity of LMC reflects different patterns of encoding for positive and negative tone commands during tone production. To definitively demonstrate that this activity reflects feed-forward control of laryngeal muscles, we used direct focal electrocortical stimulation (DES) during intraoperative clinical awake brain mapping. We asked whether stimulating the pitch-encoding region of LMC would cause pitch rising and lowering in awake participants. Five patients (two left-sided and three right-sided) who underwent awake surgery were included to map the pitch control area. A bipolar stimulator was used to stimulate the patient's LMC when the patient was pronouncing /mā/ or /má/ to map the pitch rising or pitch lowering area, respectively (Fig. 5a). In order to confirm the pitch change was caused by stimulation, the utterances of /mā/ and /má/ were also recorded as the control condition at least five times without stimulation. The positive sites for pitch change were validated at least five times.

We found that stimulation of dLMC not only evokes pitch rising[8], but also elicits pitch lowering (Fig. 5b–d). When the participant was uttering the /mā/ syllable, stimulation to the Label 1 site (Fig. 5b) of dLMC significantly increased the fundamental frequency of this utterance (Fig. 5c, the red line represents $p < 0.05$ after FDR correction). On the contrary, when the patient was pronouncing the /má/ syllable, stimulation to the Label 2 site (Fig. 5b) of the dLMC evoked a significantly lower pitch relative to the control group (Fig. 5d, the red line represents $p < 0.05$ after FDR correction). Concordant with the previous work[8], pitch rising sites could be evoked in the bilateral dLMC (Fig. 5e, f). In addition, we also elicited pitch lowering sites in the left dLMC of two patients (Fig. 5e). Together, these results provide causal evidence for the positive and negative tuning patterns of LMC. In addition to bidirectional modulation responses of pitch evoked by electrical stimulation, we also localized motor responses, speech arrest (defined as a cessation of continuous speech while retaining the movement of non-laryngeal articulators) and anomia (characterized by the inability to retrieve names while retaining the speech ability) responses. At the group level, we observed speech arrest evoked in the bilateral dLMC, specifically near the pitch modulation sites, as well as in the left vLMC (Fig. 5e, f).

## Discussion

Using high-density direct cortical recordings and cortical stimulation, we probed the neural mechanism of precise pitch control underlying lexical tone production. Our findings suggest that distributed neural populations in both the ventral and the dorsal LMC are involved in the neural coding of dynamic vocal pitch control. In particular, these neural populations encode two distinct laryngeal movements that lead to pitch rising and pitch lowering independently.

There is little knowledge about the precise neural mechanism underlying vocal pitch control. One of the major challenges is the dynamic nature of vocal pitch during the speech, where rapid pitch changes would take place within 100 ms. As a result, previous imaging studies, which are constrained by low temporal resolution, failed to reveal the fine-scale neural coding in LMC that supports the dynamic control of vocal pitch[18–21]. The high-density ECoG grid used in this study overcomes this constraint in temporal resolution while maintaining millimeter-level spatial resolution, facilitating our ability to identify distinct neural populations within LMC that contribute to different transient pitch dynamics.

Our results extend the understanding of pitch control to tonal languages. Recent human intracranial electrophysiology studies have demonstrated that dorsal LMC contributes to vocal pitch control during speech intonations in non-tonal languages[8,9,22]. However, intonations in non-tonal languages, such as English, do not exploit the full range of dynamics in pitch space[23]. In particular, most of the dynamics in the stress patterns in the intonations only include pitch rising above the neutral pitch, while in tonal languages, pitch dynamics also include frequent pitch lowering below neutral pitch, such as the dipping tone in Mandarin Chinese. As a result, although these studies identify neural populations that correlate to vocal pitch, they do not cover the full neural codes of pitch control. Here we investigate speech production in a typical tonal language to address this gap in the literature. With the combination of correlational encoding models and causal DES methods, we established a refined cortical map of fine-grain pitch control during speech production. Our results indicate that rather than a linear monotonic coding of the actual vocal pitch, there are distinct neural populations that encode the more complex movements of pitch rising and pitch lowering.

One of the key questions in understanding the neural mechanisms in sensorimotor cortex (SMC) is the level of representation in the neural coding. Previous studies have suggested that rather than coding for the movements of individual muscles, the neural activity in single neurons and local populations in SMC is correlated with coordinated movements of multiple muscles in both primates[24] and humans[25]. Here using a modified tone production model, we demonstrate that the correlation between neural activity in LMC, a specific subregion of SMC, and the pitch dynamics can be explained by the neural representation of two different types of pitch control commands, pitch rising and pitch lowering. Each of the two commands can be achieved by a coordinated movement of the CT and TA muscles[4,6,7]. The actual relationship between the laryngeal muscles and the pitch dynamics is complex and each pitch movement may require the coordinated

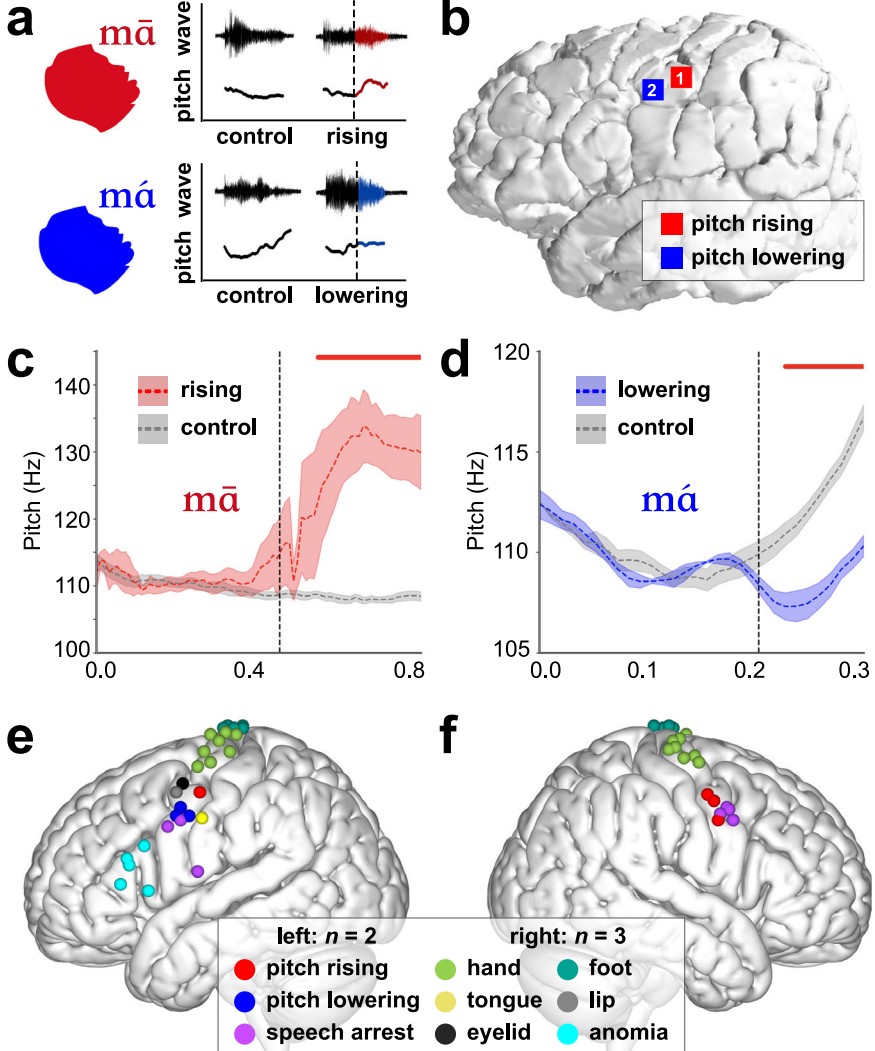

**Fig. 5 | Direct electrical cortical stimulation of dLMC elicits pitch rising and lowering. a** Acoustic waveform and absolute pitch diagrams of two syllable production paradigms. The pitch of normal /mā/ decreases slowly (control, marked in gray), and the pitch rising response is defined as the direct electrical stimulation (DES, black dotted line) evoked increase in pitch (marked in red) when the patient is pronouncing /mā/. The typical /má/ pitch drops first and then rises rapidly (control, marked in gray). The pitch lowering response is defined as the DES (black dotted line) evoked decreases in absolute pitch (marked in blue) relative to the control group. **b−d** In a typical patient, the pitch rising and pitch lowering responses were both evoked in the patient's left dLMC. **b** Cortical representation of the pitch rising (the red Label 1) and pitch lowering (the blue Label 2) sites in this typical case. **c** Average pitch contours of the electrical stimulation-induced pitch rising response at T1 (in red) compared to the average pitch contours of normal /mā/ (in gray). The horizontal dotted line and shaded area represent the mean ± standard error of the mean (s.e.m.) across repetitions. The vertical dotted line represents the average start time of electrical stimulation (mean ± s.e.m.: 452 ± 49 ms). **d** Average pitch contours of the electrical stimulation-induced pitch lowering response at T2 (in blue) compared to the average pitch contours of normal /má/ (in gray). The horizontal dotted line and shaded area represent the mean ± s.e.m. across repetitions. The vertical dotted line represents the average start time of electrical stimulation (mean ± s.e.m.: 203 ± 19 ms). The red solid line indicates time points survived $p < 0.05$, two-sided paired $t$ test, FDR correction). **e, f** The group-level cortical distributions of pitch rising (red), pitch lowering (blue), speech arrest (purple) and other sites (coded by distinct colors) on the (**e**) left and (**f**) right hemispheres. Source data are provided as a Source Data file.

contractions of multiple laryngeal muscles. In fact, it is also possible that different configurations of laryngeal muscle movements would lead to the same vocal pitch output[26,27]. Furthermore, consistent with previous DES studies, speech arrest could be induced in both the vLMC and the dLMC[28,29]. Previous research commonly interprets speech arrest as a negative motor response or a disruption in speech planning/coordination[30–32]. However, the lack of monitoring of laryngeal muscles and vocal folds prevents us from fully excluding the possibility of direct involvement of laryngeal movements in speech arrest[28,29]. To fully uncover the exact mapping between activity in these neural populations and the laryngeal muscle movements may require single neuron recording combined with electromyography monitoring and direct visualization of the vocal fold under fiberoptic bronchoscopy.

Recent advances in brain-computer interface (BCI) have shown that it is possible to decode articulator movements and generate speech from the neural activity recorded in SMC[33–36]. However, previous studies are using non-tonal language where pitch information is not critical for conveying lexical information. For tonal languages, in addition to syllables that are determined by articulator movements, the lexical pitch is also crucial for conveying word meanings. Here using multivariate pattern analysis, we show that lexical tone can be decoded from the distributed neural activity patterns in SMC. This provides a potential implementation of a speech BCI system for tonal languages like Mandarin Chinese, where articulator movements and pitch dynamics can be decoded from the distributed neural activity in SMC through different channels and combined to generate speech veridically.

To sum up, using high-density electrophysiology recordings and direct cortical stimulations, we reveal the neural mechanisms underlying precise pitch control during speech production of tonal languages, with distinct populations in the bilateral LMC contributing to the rising and lowering of vocal pitch. These findings not only extend our knowledge of the laryngeal motor cortex and its neural coding, but also indicate new applications in speech BCI for tonal languages.

Our study also carries certain limitations. Due to the notable inter-individual variability of tone production cortex and the constraint of not being able to concurrently record from both the left and right LMC within the same patient, we were unable to observe bidirectional tuning across all participants, as indicated by the findings at the population level. In forthcoming research endeavors, we plan to include a larger number of participants to gain deeper insights into the spatial distributions and individual disparities of tone production cortex.

## Methods

### Participants

A total of 8 subjects from Huashan Hospital participated in this study. All participants (age range: 29–51 years; 5 males, 3 females; 4 left, 4 right) were eloquent brain tumor patients undergoing awake language mapping as part of their surgery. During the intraoperative language mapping, high-density electrode grids were temporarily placed onto the sensory-motor cortex to record local field potentials from the cortex, and the participants were instructed to perform the experiment tasks.

Subjects were asked to participate in the research study only if they were undergoing awake surgery with direct cortical stimulation as part of normal clinical routine, meaning that this was deemed necessary for the safe resection of their tumor. Each participant was consented prior to the surgery, at which time it was explained in a transparent manner (as detailed in the IRB-approved written protocol/consent document) that the research task was for scientific purposes and would not directly impact their care. It was clearly articulated to each subject that participation in the research task was completely voluntary. The experimental protocol was approved by the Huashan Hospital Institutional Review Board of Fudan University (HIRB, KY2017–437). All participants gave their written, informed consent prior to testing.

### Experiment paradigm

Two different versions of the tone production paradigm were performed. Two subjects (S1 and S2) participated in the first paradigm, while the other subjects participated in the second paradigm. For each trial in the first version, a single word was presented on a screen 20 cm from the participant. The subject was instructed to name the word three times in a row after a go cue. In each block, a total of 60 different words were used (four lexical tones of /e/, /ye/, /yue/, /wo/, /ei/, /wei/, /ou/, /you/, /a/, /ya/, /wa/, /ai/, /wai/, /ao/, /yao/), and each word was repeated 3 times, which yielded 180 tokens in total (45 repetitions per lexical tone). Subject S1 completed one block (45 repetitions per lexical tone), and Subject S2 completed two blocks (90 repetitions per lexical tone). For each trial in the second version, a single word was played through the speaker of a laptop, and the participant was instructed to repeat the exact word three times in a row. In each block, a total of 8 different words were used (four lexical tones of /ma/ and /mi/, i.e., /mā/, /má/, /mǎ/, /mà/, /mī/, /mí/, /mǐ/, /mì/), and each word was repeated 15 times, which yielded 120 tokens in total (30 repetitions per lexical tone). Each patient completed four blocks (120 repetitions per lexical tone).

In the sentence production task, we selected 20 sentences from the Fu-sentence corpus[16], which consists of phonetically balanced 7-character Mandarin sentences (e.g., "北京近来很寒冷", /běi jīng jìn lái hěn hán lěng/). For each trial in the task, a single sentence was presented on a screen positioned 20 cm away from the participant. The participant was instructed to read the entire sentence aloud once.

Within each block, each sentence was repeated twice. Each participant completed 2-5 blocks, resulting in a total of 80-200 sentences (560-1400 tones) in total.

### Data acquisition and preprocessing

During the experimental tasks, neural signals were recorded from one or two 128-channel ECoG grids (8 × 16, 4 mm spacing) using a multi-channel amplifier optically connected to a digital signal processor (Tucker-Davis Technologies). The local field potential at each electrode contact was amplified and sampled at 3052 Hz. The raw voltage waveform was visually examined, and channels containing signal variation too low to be detectable from noise or continuous epileptiform activity were removed. Time segments on remaining channels that contained electrical or movement-related artifacts were manually marked and excluded. The signal was then notch-filtered to remove line noise (at 50 Hz, 100 Hz, and 150 Hz). Using the Hilbert transform, the envelopes of the signal outputs filtered by eight Gaussian filters (center frequencies: 70-150 Hz, log spaced) were computed. The high-gamma signal was taken as the average analytic amplitude (envelope) across these eight bands. The signal was down-sampled to 100 Hz and z-scored using the entire recording block for normalization.

### Principal component analysis (PCA) of the lexical tones

We used PCA to analyze the acoustic pitch space of lexical tones. We extracted the pitch contours of the eight tone exemplars in the tone production tasks. We utilized an interpolation-based method for proportional warping to adjust the length of these contours to a target duration of 400 ms (40 time points at 100 Hz sampling rate). PCA was performed on this 2891 × 40 data matrix $X$, and we got decomposition $X = LW^T$, where $L$ is a 2891 × 40 PC score matrix, and $W$ is a 40 × 40 orthogonal weight matrix with columns of $W$ forming an orthogonal basis set for the 40 temporal features. The minimum number of principal components (PCs) necessary is determined to achieve a cumulative explained variance ratio surpassing 95%.

### Electrode localization

To localize the electrode, the three-dimensional positions of the grid corners were recorded using the Medtronic neuronavigation system intraoperatively. These corner electrodes were then aligned to the pre-operative MRI, using intraoperative photographs as reference. In the end, we used the "img_pipe" package in Python to localize the remaining electrodes by interpolating and extrapolating from the corner electrodes[37].

### Speech-responsive electrodes selection

To find speech-responsive electrodes in the SMC and adjacent frontal cortex, we first aligned high-gamma responses to the onsets of syllable production. Onsets were defined as times where the sound of syllable production was preceded by at least 400 ms of silence. We then calculated the average high-gamma responses at each time point ($mean_{real}$) around the speech onset (time window from 300 ms before onset to 100 ms after onset). We randomly sampled 1000 time points as the onsets, and calculated the corresponding average responses ($mean_{permutation}$) and standard deviation ($SD_{permutation}$) in the same time window. If the $mean_{real}$ of continuous 100 ms on an electrode was outside the range of $mean_{permutation} \pm 5SD_{permutation}$), the electrode was considered as a speech-responsive electrode.

### Tone discriminant electrodes selection

To find speech-responsive electrodes that also discriminate between lexical tones, we first aligned high-gamma responses to the onsets of the tones. Following that, we tested whether the mean high-gamma responses were significantly different among the four tones using the one-way ANOVA. Specifically, we compute the $F$-statistic for every time point during the -300 ms to 200 ms time period relative to the tone

onset (50 total time points) and find significant time points with $p < 0.05$ threshold using Bonferroni correction for the total number of electrodes and time points. Only the electrodes with at least 10 consecutive significant time points (100 ms) were considered as tone discriminant electrodes.

## Speech feature extraction

In line with our previous research, we employed a similar Praat parameter extraction approach[8,14,38]. We extracted the pitch contour of each syllable with an autocorrelation method in Praat (Version 6.1.01, https://www.fon.hum.uva.nl/praat/)[39]. Additionally, we addressed halving and doubling errors during the extraction process. Individual pitch minimum and maximum values were determined for each participant, and a timestep of 0.01 s was employed. All other parameters adhered to the default settings of Praat. The intensity was also extracted from each trial using Praat and normalized (z-score) within each block. Subsequently, we calculated pitch height, pitch change, and binary pitch based on the original pitch contour.

Specifically, assume that the absolute pitch height value at any given time $t$ was $F_0(t)$. $F_0(t)$ could be a positive number if at that time the phoneme was voiced, or NaN if not voiced. The pitch height ($h$) was computed as the logarithm of the absolute pitch frequency $h(t) = \log p(t)$ (log Hz, or octave); pitch change ($c$) was the first-order difference of the log pitch height $c(t) = h(t) - h(t-1)$; binary pitch $b(t) = 1$ (if $h(t)$ is not NaN) and $b(t) = 0$ (if $h(t)$ is NaN). Furthermore, we discretized absolute pitch, relative pitch and pitch change into 10 bins, equally spaced from the 2.5 percentile to the 97.5 percentile value. The bottom and top 2.5% of the values were placed into the bottom and top bins respectively. As a result, pitch height and pitch change were both represented as 10-dimensional binary feature vectors. For non-pitch periods, these feature vectors would have all 0 s.

In addition to pitch-related features, we also included 'non-laryngeal articulator movements' as another binary variable that summarizes the differences between the consonants and the vowels[9], which are mainly articulatory differences in jaw, lips and tongue.

## Fujisaki model

Using the Fujisaki model of vocal pitch, the pitch contour of an utterance ($F_O$) can be given by the following equations:

$$\ln F_0(t) = \ln F_b + P + T \tag{1}$$

$$P = \sum_{i=0}^{I} A_{p,i} G_p(t - T_{0i}) \tag{2}$$

$$T = \sum_{j=1}^{J} A_{t,j} \left\{ G_t(t - T_{1j}) - G_t(t - T_{2j}) \right\} \tag{3}$$

$$G_p(t) = \begin{cases} \alpha^2 t e^{-\alpha t}, t \geq 0, \\ 0, t < 0 \end{cases} \tag{4}$$

$$G_t(t) = \begin{cases} min[1 - (1 + \beta t)e^{-\beta t}, \gamma], t \geq 0, \\ 0, t < 0 \end{cases} \tag{5}$$

In the equations, $F_b$ represents baseline value of fundamental frequency. The phrase component ($P$) describes the slow declination in pitch over the course of a phrase[40], which consists of $I$ individual phrase commands of amplitude $A_{p,i}$ and shape $G_p$. The tone component ($T$) consists of $J$ individual tone commands of amplitude $A_{t,j}$ and shape $G_t$, which describes the bidirectional control of vocal fold length and tension by rapid movements of the laryngeal muscles, resulting in

bidirectional pitch modulation. $G_p$ is an impulse response function used to describe the phrase control mechanism, while $G_t$ is a step response function used to describe the tone control mechanism[17]. It should be noted that in the process of single syllable production, the influence of the phrase component was not considered, and the $P$ was set to 0. As a result, the first equation can be simplified as: $\ln F_0(t) = \ln F_b + T$.

FujiParaEditor was used to estimate the phrase and tone components for each syllable and spoken sentence[41]. An automated inference process was used[42], as well as manual corrections when necessary.

## Encoding model

We used time-delayed linear encoding models, known as temporal receptive field models[43], to evaluate what features are driving the neural activity in LMC during lexical tone production. Temporal receptive field (TRF) models predict neural activity using speech-related features in a window of time around the neural activity. In particular, we fit the linear model $y(t) = \sum_{f=1}^{F} \sum_{\tau=0}^{T} \boldsymbol{\beta}_f^T(\tau) \boldsymbol{x}_f(t - \tau) + \epsilon$ for each electrode, where $y$ is the high-gamma activity recorded from the electrode, $\boldsymbol{x}_f(t - \tau)$ is the stimulus representation vector of feature set $f$ at time $t - \tau$, $\boldsymbol{\beta}_f(\tau)$ is the regression weights for feature set $f$ at time lag $\tau$, and $\epsilon$ is the gaussian noise.

In the full TRF model, we included features for the sound intensity, relative pitch height, pitch change, syllable onset, binary pitch, tone command and tone category. To calculate the unique contribution of specific features, we fit TRF models that excluded each feature in turn and calculated the difference in $R^2$ between the full and reduced models.

To prevent model overfitting, we used L2 regularization and cross-validation. Specifically, we divided the data into three mutually exclusive sets of 80%, 10% and 10% of samples. The first set of 80% was used as the training set. The second set was used to optimize the L2 regularization hyperparameter, and the final set was used as the test set. We evaluated the models using the correlation between actual and predicted values of neural activity on held out data. We performed this procedure 5 times and the performance of the model was taken as the mean of performance across all testing sets.

To calculate the significance of unique portions of variance explained, we employed permutation testing. We shuffled the acoustic features between all syllables in the stimuli before computing null values of the unique variance explained by each feature by running the same analysis pipeline. We ran this procedure 200 times to get a null distribution of values. Using this empirical null distribution, we determine a significant threshold of $p < 0.005$ for each unique $R^2$ value.

## Analysis of tuning curve

To evaluate the functional relationship between the tone commands and the neural activity in LMC, we computed the tuning curve for each speech-responsive LMC electrode. The tone commands were discretized into 12 bins uniformly spanning the middle 99-percentile range for each individual subject. The mean and standard deviation of high-gamma activity in each electrode was calculated for each bin to get the entire tuning curve.

Specifically, the tone commands value at time $t$ is denoted as $A(t)$. We estimated the distribution of $A(t)$ as CDF($A$) for each subject, using the histogram of $A(t)$ over all time $t$. Then according to the quantiles $q_i$ of this histogram $\{q_i = CDF^{-1}((i-1)/12 + 0.005)|i = 1, \ldots, 13\}$, we can sort the value into 12 bins. As a result, $A(t) = i$, if $q_i < A(t) \leq q_{i+1}$. Since HGA (denoted as $h(t)$) and tone command ($A(t)$) were synchronously recorded, so we had paired $A(t)$ and $h(t)$. Then for the $i$-th bin of the tone command, we collected set as $H_i = \{A(t)|\forall t, s.t. A(t) = i\}$, and compute the mean and standard deviation of this set $H_i$. Then the tuning curve can be plotted as $x = i$ and $y = $ mean ($H_i$). A linear model $y = ax + b$ was fitted between the tone commands ($x$) and mean high-gamma ($y$) in the tuning curve using ordinary least squares. We used the slope $a$ to determine the polarity of the tuning curve.

## Population tone decoding

To determine whether neural responses to different tones were distinguishable in the LMC network, we used multivariate pattern analysis to evaluate the tone decoding accuracy in LMC neural population. We aligned high-gamma responses to the onsets of the vowels of syllables[9] and divided trials by the lexical tone of the syllable. A sliding time window with a length of 50 ms (5 consecutive time points of high-gamma activity) was used to evaluate the dynamics of neural representation. For each sliding window, the neural activity across all speech-responsive electrodes was concatenated and used as features to train a pattern classifier. The time course of the averaged pairwise classification accuracy was computed as the final decoding result. Specifically, we used logistic regression with L2 penalty across electrodes. This approach would avoid overfitting while maintaining temporal smoothness and network interactions between local populations. A nested cross-validation strategy was adopted where 5-fold cross-validation was used to estimate the classification accuracy, and within each training set, 10-fold cross-validation was used to select the optimal penalty parameter $\lambda$.

## Intraoperative direct electrical stimulation mapping

Five consecutive glioma patients (two left-sided and three right-sided) who underwent awake surgery at Huashan Hospital between August 2020 to January 2021 were included to map the pitch control area. We used the same stimulation parameters (5-mm interval, bipolar electrode, current-constant bipolar square wave, 1-ms wave width, and 60 Hz frequency, 1-3 mA intensity) as our previous study[29,30]. All patients are all (1) right-handed, (2) had intact consciousness, language function, and motor function before the operation, and (3) the ventral central lobe was completely exposed during the operation. During intraoperative direct electrical stimulation mapping, the cortex was stimulated at a 1 cm interval. Prior to mapping the pitch control area, we conducted mapping of orofacial and limb muscle movements, speech arrest, and anomia, as extensively documented in our previous studies[29,30]. Motor responses were defined as visible muscle movements or observable motor waveforms on electromyography (EMG). Speech arrest was defined as a complete cessation of ongoing speech during a counting task while retaining consciousness and the ability to move non-laryngeal articulator muscles. Anomia was defined as the inability to name the presented objects in a picture naming task but still being able to generate the introductory phrase "This is a...". We first stimulated the motor area and increased the current intensity until the motor responses were evoked. Then, the same current intensity was applied to locate the pitch control area[29]. Two syllable paradigms were used to map pitch rising and lowering regions respectively. Typically, the pitch of the normal /mā/ syllable decreases slowly, and the pitch rising response is defined as a DES-evoked increase in pitch when the patient is pronouncing /mā/ (Fig. 5a). The pitch of the typical /má/ drops first and then rises rapidly. The pitch lowering response is defined as the DES-evoked decrease in absolute pitch relative to the control group (Fig. 5a). In each task, the normal syllable was repeated at least five times. Each cortical site was discontinuously stimulated in the middle of the patient's syllable pronunciation (Fig. 5a) at least three times and would be determined as a positive site if at least two-thirds of them evoked pitch change without after-discharge. The positive sites were marked with sterile labels. Finally, the intraoperative photos, neuro-navigation snapshots, surgery video under the microscope and the synchronized audio were recorded through our Brain Mapping Interactive Stimulation System[44].

After the operation, we used Praat software (Version 6.1.01, https://www.fon.hum.uva.nl/praat/) to extract and verify the difference in pitch contours between the control group and the DES group (two-tailed paired *t* test, $p < 0.05$ after FDR correction was considered significant). In addition, to explore the spatial distribution of the pitch control area, we reconstructed the brain surface of each patient and manually normalized the pitch rising and lowering sites to the ICBM 152 template based on the anatomical landmarks.

## Reporting summary

Further information on research design is available in the Nature Portfolio Reporting Summary linked to this article.

## Data availability

The human patient data relevant to this study are accessible under restricted access according to our IRB protocol. The de-identified patient data that support the findings of this study will be made available from the corresponding author upon request. Source data are provided with this paper.

## Code availability

The completely developed code that operates on the full data set will be made available from the authors upon reasonable request.

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

## Acknowledgements

The authors would like to thank Prof. Matthew Leonard for helping to design the linguistics task, Jinxuan Wu for drawing the anatomy illustration in Fig. 1, and Prof. Hansjörg Mixdorff and Dr. Kaile Zhang for the instructions on generating tone commands using FujiParaEditor and Pratt. Dr. Junfeng Lu is supported by STI 2030—Major Projects (2022ZD0212300), Shanghai Pujiang Program (21PJD007). Dr. Jinsong Wu and Ying Mao receive funding from Shanghai Municipal Science and Technology Major Project (2018SHZDZX01). Dr. Jinsong Wu receives funding from Innovation Program of Shanghai Municipal Education Commission (2023ZKZD13). Dr. Yuanning Li is supported by Shanghai Pujiang Program (22PJ1410500) and the National Natural Science Foundation of China General Program (32371154). Dr. Edward Chang is supported by funding from NIH grant U01 NS117765.

## Author contributions

Conception and design of the work, E.F.C., J.W., J.L., Y. Li, and Z.Z.; acquisition of data, J.L., Y. Li, Z.Z., Y. Liu, Y.Z.; analysis and interpretation of data, J.L., Y. Li, and Z.Z.; writing—original draft, J.L., Y. Li, and Z.Z.; writing—review & editing, E.F.C., J.W., J.L., Y. Li, and Z.Z.; resources: Y.M., J.W., and E.F.C.; supervision: Y.M., J.W., and E.F.C.

## Competing interests
