## [Peer Review File · Nature Communications]

Neural Control of Lexical Tone Production in Human Laryngeal Motor CortexEditorial Note: This manuscript has been previously reviewed at another journal that is not operating a transparent peer review scheme. This document only contains reviewer comments and rebuttal letters for versions considered at Nature Communications.

Reviewers' Comments:

Reviewer #1:

Remarks to the Author:

The submitted manuscript "Neural Control of Lexical Tone Production in Human Laryngeal Motor Cortex" by Lu and colleagues seeks to investigate the neural basis of the motor control of lexical tone in tonal languages. The authors indicate that rather than tone category selective coding, they find that tones are encoded through articulatory kinematic information. They further use a computational model of tone production to disassociate two elements of population neural activity for tone generation, and they also use direct electric stimulation to assess the causality of these sites.

Overall, it is an interesting manuscript (particularly the stimulation component), and one which I have previously reviewed for a different journal and I am pleased to say that the authors have largely addressed my previous concerns. I have only two outstanding comments.

1. It would be useful to show the tumor and exposure (craniotomy) brain maps to assess both tumor extent and eloquent cortex in relation to electrode coverage. If PHI is a concern, figures outlining the craniotomy/tumor in relation to the electrode coverage would suffice. For instance, a figure similar to Figure S5 but with the addition of the craniotomy boundary/tumor location relative to the array.
2. Typo: Page 4 Line 74 Should read "A previous intracranial study..."

Reviewer #3:

Remarks to the Author:

The reviewers have addressed most of my concerns. I still have one point of contention regarding my original major point 2: "More importantly the tuning is not robust on cortex, if you compare the distribution across participants in Figure S5 the pattern is quite variable (with no tuning for patient S6 and only one directional tuning in patients S2,S3,S4,S7). The population response in Figure 3 represents both directions but this really does not replicate for the majority for your patients." The authors have responded explaining why some participants exhibited unidirectional tuning on one side only and that based on the group-level patterns of high-gamma activity they believe that there are two distinct commands in the brain, resulting in four different tones. Overall, I agree however there is quite a discrepancy between the gist of the results as presented in Figure 3 vs. the within subject data in S5. The interpretation of a population response across this limited number of patients (with limited tuning) has to be explicitly stated and discussed as a limitation. I would request:

1. Clear statements in the manuscript as to the inter subject variability.
2. The authors update the units in S5 (they still appear as min/max) as they did in the rest of the manuscript so units can be compared across Figures S5 and 3.
3. Some of the electrodes within subject (patient S3) are not clearly visible in Fig 3e, perhaps due to the MNI transformation plus angle of view. However, I think it would be helpful to add a depiction of all positive and negative electrodes in MNI space (akin to 3e) but color coded/denoted by patient so the spatial contribution of each patient to the population responses are clear. Likely as an addition to S5.

Dear Reviewers,

We would like to express our gratitude for your correspondence and the insightful feedback provided by the reviewers regarding our manuscript titled "Neural Control of Lexical Tone Production in Human Laryngeal Motor Cortex" (Manuscript ID: NCOMMS-23-18276-T). These comments have proven to be invaluable and exceptionally constructive. Please see our point-to-point responses below, highlighted in Arial font with left indentation. The original comments are in Times New Roman, italic font. The revised part of the manuscript is highlighted in blue font.

Reviewers' Comments:

Reviewer #1 (Remarks to the Author):

The submitted manuscript "Neural Control of Lexical Tone Production in Human Laryngeal Motor Cortex" by Lu and colleagues seeks to investigate the neural basis of the motor control of lexical tone in tonal languages. The authors indicate that rather than tone category selective coding, they find that tones are encoded through articulatory kinematic information. They further use a computational model of tone production to disassociate two elements of population neural activity for tone generation, and they also use direct electric stimulation to assess the causality of these sites.

Overall, it is an interesting manuscript (particularly the stimulation component), and one which I have previously reviewed for a different journal and I am pleased to say that the authors have largely addressed my previous concerns. I have only two outstanding comments.

We greatly appreciate Reviewer 1's positive feedback on our manuscript, especially regarding the stimulation component. Below, we have listed the point-to-point responses.

1. It would be useful to show the tumor and exposure (craniotomy) brain maps to assess both tumor extent and eloquent cortex in relation to electrode coverage. If PHI is a concern, figures outlining the craniotomy/tumor in relation to the electrode coverage would suffice. For instance, a figure similar to Figure S5 but with the addition of the craniotomy boundary/tumor location relative to the array.

We thank Reviewer 1 for the valuable suggestions. Considering PHI, we have revised Figure S2 to present panels outlining the tumor locations in relation to the electrode coverages.

Edits:

Figure S2 in Supplementary Material (Page 3):

Figure S2. Speech responsive electrodes for all participants (S1-S8). Red dots indicate speech responsive electrodes. Green regions represent the superficial tumor locations. Black dotted lines illustrate the surface projections of the deep insular and medial temporal lobe tumors.

2. Typo: Page 4 Line 74 Should read “A previous intracranial study...”

We have corrected the typo.

Edits:

Pages 3-4, Lines 73-74: A previous intracranial study¹⁴ in Mandarin tone perception demonstrated the existence of single-electrode-level high gamma responses that differentiate between lexical tones in the temporal lobe during speech perception.

Reviewer #3 (Remarks to the Author):

The reviewers have addressed most of my concerns. I still have one point of contention regarding my original major point 2: “More importantly the tuning is not robust on cortex, if you compare the distribution across participants in Figure S5 the pattern is quite variable (with no tuning for patient S6 and only one directional tuning in patients S2,S3,S4,S7). The population response in Figure 3 represents both directions but this really

does not replicate for the majority for your patients.” The authors have responded explaining why some participants exhibited unidirectional tuning on one side only and that based on the group-level patterns of high-gamma activity they believe that there are two distinct commands in the brain, resulting in four different tones. Overall, I agree however there is quite a discrepancy between the gist of the results as presented in Figure 3 vs. the within subject data in S5. The interpretation of a population response across this limited number of patients (with limited tuning) has to be explicitly stated and discussed as a limitation. I would request:

1. Clear statements in the manuscript as to the inter subject variability.

We sincerely appreciate Reviewer 3’s thorough review and constructive feedback on our manuscript. We have added clear statements regarding the inter subject variability of tone tuning in the limitations.

Edits:

Pages 10-11, Lines 297-302: Our study also carries certain limitations. Due to the notable inter-individual variability of tone production cortex and the constraint of not being able to concurrently record from both the left and right LMC within the same patient, we were unable to observe bidirectional tuning across all participants, as indicated by the findings at the population level. In forthcoming research endeavors, we plan to include a larger number of participants to gain deeper insights into the spatial distributions and individual disparities of tone production cortex.

2. The authors update the units in S5 (they still appear as min/max) as they did in the rest of the manuscript so units can be compared across Figures S5 and 3.

We have updated the units in Figure S5.

Edits:

Figure S5 in Supplementary Material (Page 7)

Figure S5. Positive and negative tuning electrodes for all participants (S1-S8), Related to Figure 3. (a) Red dots indicate positive tuning electrodes. Blue dots indicate negative tuning electrodes. Darker color indicates stronger tuning. **(b)** Group-level distribution of positive tuning electrodes on the cortical surface (color coded by subject). **(c)** Group-level distribution of negative tuning electrodes on the cortical surface (color coded by subject).

3. Some of the electrodes within subject (patient S3) are not clearly visible in Fig 3e, perhaps due to the MNI transformation plus angle of view. However, I think it would be helpful to add a depiction of all positive and negative electrodes in MNI space (akin to 3e) but color coded/denoted by patient so the spatial contribution of each patient to the population responses are clear. Likely as an addition to S5.

We have added a panel displaying all positive and negative electrodes in MNI space, color coded by patient.

Edits:

Figure S5 in Supplementary Material (Page 7)

Figure S5. Positive and negative tuning electrodes for all participants (S1-S8), Related to Figure 3. (a) Red dots indicate positive tuning electrodes. Blue dots indicate negative tuning electrodes. Darker color indicates stronger tuning. **(b)** Group-level distribution of positive tuning electrodes on the cortical surface (color coded by subject). **(c)** Group-level distribution of negative tuning electrodes on the cortical surface (color coded by subject).